# Breast-Specific Epigenetic Regulation of DeltaNp73 and Its Role in DNA-Damage-Response of *BRCA1*-Mutated Human Mammary Epithelial Cells

**DOI:** 10.3390/cancers12092367

**Published:** 2020-08-21

**Authors:** Ayelet Avraham, Susanna Feldman, Sean Soonweng Cho, Ayala Kol, Lior Heler, Emmanuela Riklin-Nahmias, Avishay Sella, Tamar Karni, Tanir M. Allweis, Saraswati Sukumar, Ella Evron

**Affiliations:** 1Department of Oncology, Shamir Medical Center affiliated to Sackler School of Medicine, Tel Aviv University, Beer-Yaakov 7033001, Israel; susannaf@shamir.gov.il (S.F.); Ayalak1977@gmail.com (A.K.); Dr.AvishaySella@shamir.gov.il (A.S.); 2Department of Oncology, Johns Hopkins University School of Medicine, Baltimore, MD 21231, USA; soonwengcho@gmail.com (S.S.C.); saras@jhmi.edu (S.S.); 3Department of Plastic Surgery, Shamir Medical Center and Sackler School of Medicine, Tel Aviv University, Beer-Yaakov 7033001, Israel; LiorH@shamir.gov.il; 4Department of Pathology, Shamir Medical Center and Sackler School of Medicine, Tel Aviv University, Beer-Yaakov 7033001, Israel; emmanuelan@shamir.gov.il; 5Department of Surgery, Shamir Medical Center and the Sackler School of Medicine, Tel Aviv University, Beer-Yaakov 7033001, Israel; TamiK@shamir.gov.il; 6Department of General Surgery, Kaplan Medical Center, Rehovot and the Faculty of Medicine, The Hebrew University, Jerusalem 7661041, Israel; tanirallweis@gmail.com; 7Department of Oncology, Kaplan Medical Center, Rehovot and the Faculty of Medicine, The Hebrew University, Jerusalem 7661041, Israel; allaev@clalit.org.il

**Keywords:** *BRCA*-mutation, cancer susceptibility, deltaNp73, DNA methylation, tissue-specificity

## Abstract

The function of BRCA1/2 proteins is essential for maintaining genomic integrity in all cell types. However, why women who carry deleterious germline mutations in *BRCA* face an extremely high risk of developing breast and ovarian cancers specifically has remained an enigma. We propose that breast-specific epigenetic modifications, which regulate tissue differentiation, could team up with BRCA deficiency and affect tissue susceptibility to cancer. In earlier work, we compared genome-wide methylation profiles of various normal epithelial tissues and identified breast-specific methylated gene promoter regions. Here, we focused on deltaNp73, the truncated isoform of p73, which possesses antiapoptotic and pro-oncogenic functions. We showed that the promoter of *deltaNp73* is unmethylated in normal human breast epithelium and methylated in various other normal epithelial tissues and cell types. Accordingly, deltaNp73 was markedly induced by DNA damage in human mammary epithelial cells (HMECs) but not in other epithelial cell types. Moreover, the induction of deltaNp73 protected HMECs from DNA damage-induced cell death, and this effect was more substantial in HMECs from *BRCA1* mutation carriers. Notably, when BRCA1 was knocked down in MCF10A, a non-malignant breast epithelial cell line, both deltaNp73 induction and its protective effect from cell death were augmented upon DNA damage. Interestingly, deltaNp73 induction also resulted in inhibition of BRCA1 and BRCA2 expression following DNA damage. In conclusion, breast-specific induction of deltaNp73 promotes survival of BRCA1-deficient mammary epithelial cells upon DNA damage. This might result in the accumulation of genomic alterations and allow the outgrowth of breast cancers. These findings indicate deltaNp73 as a potential modifier of breast cancer susceptibility in *BRCA1* mutation carriers and may stimulate novel strategies of prevention and treatment for these high-risk women.

## 1. Introduction

Women who harbor a heterozygous germline mutation in *BRCA1/2* face an extremely high risk of breast and ovarian cancer but rarely that of other cancers [1]. The BRCA1/2 proteins play a prominent role in DNA repair by homologous recombination, cell cycle control, and maintenance of genome integrity in all cells [2]. It is, however, not clear why disruption of this universal mechanism predisposes to cancer almost exclusively in the breast and ovary. Several hypotheses have been proposed to explain this phenomenon [3,4]. It has been suggested that tissue-specific features affecting differentiation and transcriptional regulation [5,6], or the local effect of estrogen and other secreted factors [7,8], may team up with BRCA deficiency to support carcinogenesis. Likewise, BRCA-deficient tumors may arise in tissues that support prolonged survival of cells that have inactivated both *BRCA* alleles, providing a window for accumulation of additional mutations and subsequent tumor outgrowth [2,9]. Yet, the body of evidence to date does not fully explain *BRCA*-related tissue-specific cancer susceptibility [10,11,12]. 

Given that DNA methylation regulates normal tissue differentiation and carcinogenesis [13], we proposed that breast-specific DNA methylation might regulate the expression of malignant transformation modulators in BRCA1 haplo-insufficient mammary epithelial cells. To address this question, we previously analyzed genome-wide DNA methylation profiles of various normal human epithelial tissues and identified gene promoters that had distinct methylation patterns in the breast as compared to other epithelial tissues [14]. One prominent gene was *TP73*, a member of the p53-related family that includes the p53, p63, and p73 proteins [15]. The *TP73* gene contains two promoters that drive the expression of two major isoform groups [16]. The P1 promoter upstream to exon 1 controls the expression of the full-length TAp73 that displays proapoptotic and tumor suppressor activity [17] and contributes to genome stability [18]. An alternative internal promoter P2, located within intron 3 of the gene, controls the expression of deltaNp73—a truncated variant lacking the transactivation domain. The deltaNp73 isoform acts as a dominant negative inhibitor of TAp73 and p53 by competing for the same DNA binding sites [19] and counteracting growth suppression [16]. Both variants of p73 play a role in DNA damage response, where TAp73, like p53, directs cell cycle arrest and apoptosis and deltaNp73 counteracts this activity and supports neoplastic transformation, cancer progression, and metastasis [20,21,22,23].

Here, we show that the *deltaNp73* promoter is unmethylated and the gene is markedly induced by DNA damage specifically in normal breast epithelium. We further link deltaNp73 induction to DNA damage response in BRCA1-deficient mammary epithelial cells.

## 2. Results

### 2.1. Tissue-Specific Hypomethylation at the deltaNp73 Promoter in Mammary Epithelial Cells

We previously compared genome-wide methylation profiles of various normal epithelial tissues by Illumina methylation array and identified differentially methylated regions (DMRs) at 110 gene promoter loci, which were specific to breast epithelium [14]. One of these breast-specific DMRs was located at the internal promoter of the *TP73* gene, which controls the expression of the N-terminus truncated variant *deltaNp73* (illustrated in Appendix A). Analysis of the array results at this locus revealed that the *deltaNp73* promoter (P2) was hypomethylated (50%) in breast compared to normal endometrial, lung, and colon epithelium as well as peripheral white blood cells. Conversely, the *TAp73* promoter (P1) which controls the expression of the full-length *TP73* gene was unmethylated in all tissues (Figure 1A). Quantitative methylation-specific PCR (q-MSP) confirmed the methylation array results in additional tissues (Figure 1B, left). To avoid the effects of tissue heterogeneity, we also analyzed promoter methylation in purified normal epithelial cells. Notably, *deltaNp73* promoter (P2) was unmethylated in human mammary epithelial cells (HMECs), whereas moderate (10−15%) or high methylation (75−90%) was noted in epithelial cells of other tissues (Figure 1B right). Moreover, *deltaNp73* (P2) was hypermethylated (75−100%) in matched fibroblasts isolated from the same breast, whereas hypermethylation (80−90%) was observed in both epithelial cells and fibroblasts isolated from the same endometrial tissue (Figure 1C). Analysis of the Encyclopedia of DNA Elements/ Hudson Alpha Institute for Biotechnology (ENCODE/HAIB) Illumina 450 K methylation array database supported these findings in additional types of purified epithelial cells (Figure 1D). Thus, the *deltaNp73* promoter region is methylated in a large collection of normal human epithelial cell types, whereas it remains exclusively unmethylated in normal breast epithelial cells.

### 2.2. DeltaNp73 Expression Was Markedly Induced by DNA Damage in Normal HMECs but Not in Other Epithelial Cell Types

We then compared deltaNp73 expression in relation to its tissue-specific promoter methylation in the various epithelial cell types. In general, basal levels of deltaNp73 mRNA were relatively low in all purified normal epithelial cells tested. Nevertheless, they were at least 25-times higher in HMECs (Figure 2A,B, data points ‘0’). Furthermore, when HMECs were exposed to DNA damaging agents (cisplatin, doxorubicin), the expression of deltaNp73 was markedly induced, whereas moderate or no induction was noted in the other epithelial cell types (Figure 2A,B). Analysis of deltaNp73 protein in HMECs treated with cisplatin confirmed mRNA results (Figure 2C). Of note, cell cytotoxicity by cisplatin or doxorubicin differed between cell types and did not correlate with deltaNp73 induction (Appendix A). Thus, the induction of deltaNp73 by DNA damage in the various cell types negatively correlated with *deltaNp73* promoter methylation. In addition, other major DNA damage responsive genes including *PUMA*, *p21*, and *NOXA* did not exhibit a unique expression pattern in breast epithelial cells when subjected to similar DNA damaging stimuli (Appendix A). Altogether, these findings indicate that tissue-specific promoter DNA methylation regulates deltaNp73 expression. Accordingly, expression of deltaNp73 markedly increased by DNA damage in HMECs, but not in other types of epithelial cells. Moreover, the basal expression of deltaNp73 and its induction following exposure to doxorubicin and cisplatin was much greater in HMECs than in fibroblasts from the same breast tissue (Figure 2D), in line with *deltaNp73* promoter methylation in fibroblasts (Figure 1C). In addition, the relative expression of deltaNp73 to TAp73, which affects cell viability and tumorigenic potential in DNA damage response [23], was greater in HMECs as compared to other normal epithelial cells following DNA damage (Figure 2E). This suggests that tissue-specific induction of deltaNp73 and increased ratio of deltaNp73 to TAp73 might enhance antiapoptotic pathways during DNA damage response specifically in HMECs.

### 2.3. DeltaNp73 Protected HMECs from Cell Death Following Exposure to Cisplatin 

We then studied the antiapoptotic effect of deltaNp73 in HMECs following DNA damage. Furthermore, to uncover a possible cross effect between BRCA1 and deltaNp73, both participating in DNA damage response, we also tested HMECs from *BRCA1* mutation carriers. Consistent with its tissue-specific pattern of methylation, the *deltaNp73* promoter was unmethylated in HMECs and fully methylated in WBCs of both *BRCA1*-wild type and *BRCA1*-mutant cells (Appendix A). In line with that, deltaNp73 was induced by DNA damage in both HMEC subgroups (Figure 3A, siControl). We, therefore, knocked down deltaNp73 by two different gene-specific siRNAs (siΔNp73#1 and #2), which inhibited gene expression (mRNA) by 77% (61–87%) relative to siControl (Figure 3A). We then exposed the cells to DNA damage and measured cell viability. Of note, in line with previous knowledge, HMECs-*BRCA1*-mut were more sensitive to cisplatin as compared to HMECs-*BRCA1*-wt (Appendix A). Furthermore, inhibition of deltaNp73 increased cell death in response to cisplatin in both *BRCA1*-wt (Figure 3B, ** *p* = 0.0085) and *BRCA1*-mut HMECs (Figure 3C, *** *p* = 0.0004). The comparison between the siControl and sideltaNp73 curves was done by linear mixed model analysis (Appendix A). Interestingly, similar knockdown of deltaNp73 did not significantly affect cell death when DNA damage was induced by doxorubicin (Figure 3D,E). Possibly, the role of deltaNp73 differs between DNA damage responses elicited by these two cytotoxic agents. In summary, these findings indicate that induction of deltaNp73 by cisplatin protects both *BRCA1*-wt and *BRCA1*-mut HMECs from ensuing cell death. 

### 2.4. DeltaNp73 Expression and Antiapoptotic Activity Following DNA Damage Were Augmented in BRCA1 Knocked-Down MCF10A Cells 

To elaborate on the role of deltaNp73 in BRCA-deficient breast epithelial cells, we then stably knocked down BRCA1 by two different shRNAs (shBRCA1#1, #2) in MCF10A, an immortal non-malignant breast epithelial cell line (Figure 4A). Like in HMECs, the *deltaNp73* promoter (P2) was unmethylated in MCF10A and the gene was markedly induced by various DNA damaging signals, including doxorubicin, cisplatin, and ionizing irradiation (Appendix A). Notably, induction of deltaNp73 (mRNA) by DNA damage was augmented in the BRCA1-KD MCF10A cells as compared to their BRCA1-wild type counterparts. Moreover, in line with deltaNp73 negative regulation of TAp73, this was associated with reciprocal repression of the full-length variant in BRCA1 deficient cells (Figure 4B, cisplatin and Appendix A, doxorubicin). These results were confirmed by protein analysis (Western blot) that showed 1.6–2-fold increase in deltaNp73 in the BRCA1-KD cells compared to parental MCF10A, following induction with cisplatin (Figure 4C). As in HMECs, knockdown of deltaNp73 by siRNA resulted in increased MCF10A cell death in response to cisplatin (Figure 4D, Appendix A). Moreover, the protective effect of deltaNp73 was more pronounced in BRCA1-KD cells (Figure 4E). Thus, the differences between siControl and sidNp73 death-curves, compared with the linear mixed model (Appendix A), were * *p* = 0.0379 for parental MCF10A and *** *p* = 0.000078 and *** *p* = 0.000141 for BRCA1-KD cells. Similar results were observed with doxorubicin (Appendix A). In summary, following DNA damage, both deltaNp73 induction and its protective effect from ensuing cell death were augmented in BRCA1-deficient MCF10A. These findings may suggest a possible link between BRCA1 and deltaNp73 through DNA damage response. 

### 2.5. Overexpression of DeltaNp73 Protected BRCA1-Deficient MCF10A Cells from DNA-Damage-Induced Cell Death 

To substantiate the findings obtained by deltaNp73 knockdown, we stably overexpressed Flag-deltaNp73 in BRCA1-wt and BRCA1-KD MCF10A, under the control of a doxycycline-inducible promoter. Following induction with doxycycline, Flag-deltaNp73 mRNA and protein were clearly overexpressed (Figure 5A). Corresponding expression of BRCA1 in these clones is shown in Appendix A. Notably, induction of exogenous deltaNp73 resulted in 100-fold more mRNA (Figure 5B) and 23-fold more protein (Figure 5C) as compared with endogenous deltaNp73 induced by 15 μM cisplatin. We then induced Flag-deltaNp73 and exposed the cells to increasing doses of cisplatin simultaneously. Consequently, cell viability increased in BRCA1-KD, whereas the BRCA1-wt cells were unaffected (Figure 5D–F). Of note, as in *BRCA1*-mut HMECs and in line with the literature, the BRCA1-KD MCF10A clones were more sensitive to cisplatin than the BRCA1-wt MCF10A clone, but the opposite was true when deltaNp73 was overexpressed with doxycycline (Appendix A). These results suggest that deltaNp73 has a more profound effect in protecting cells with BRCA1-deficiency from DNA damage-induced cell-death.

### 2.6. Induction of DeltaNp73 by DNA Damage Was Associated with Down-Regulation of BRCA1 and BRCA2 in Breast Epithelial Cells

In view of greater induction of deltaNp73 in BRCA1-KD MCF10A cells compared to BRCA1-wt (Figure 4B,C), we looked for possible interplay between deltaNp73 and BRCA1 upon DNA damage. Interestingly, induction of deltaNp73 by various DNA damaging stimuli (cisplatin, doxorubicin or ionizing irradiation) coincided with reduction in BRCA1 and BRCA2 expression (mRNA) in MCF10A (Figure 6A) and in HMECs (Appendix A). Furthermore, overexpression of exogenous deltaNp73 in MCF10A led to successive reduction in BRCA1 and BRCA2 (Figure 6B). In line with mRNA, induction of deltaNp73 protein (6.6-fold) corresponded with 70% reduction in BRCA1 protein in cisplatin-treated MCF10A compared with non-treated cells (Figure 6C, left), and overexpression of exogenous deltaNp73 was associated with 30% reduction in the BRCA1 protein (Figure 6C, right). Similar induction of deltaNp73 and reduction in BRCA1 proteins was observed in HMECs following treatment with cisplatin (Appendix A). Accordingly, knockdown of deltaNp73 in MCF10A resulted in increased BRCA1 and BRCA2 mRNA, both in non-treated and in cisplatin-treated cells (Figure 6D). Altogether, these findings suggest a feedback regulatory loop of deltaNp73 and BRCA1expression upon DNA damage. Thus, induction of deltaNp73 was associated with reduction in BRCA1/2 (Figure 6), whereas reduction in BRCA1 associated with increase in deltaNp73 expression (Figure 4B).

### 2.7. Hypermethylation of the DeltaNp73 Promoter in Breast Cancers

In view of our data showing breast-specific DNA methylation at the *deltaNp73* promoter in normal epithelial tissues, we further studied *deltaNp73* promoter methylation in breast tumors. Survey of the cancer genome atlas (TCGA) revealed hypermethylation across the entire TP73 gene in breast cancers, as compared to normal breast tissues, that varied by estrogen receptor (ER) status and intrinsic mRNA (PAM50) subtypes [24] (Appendix A). In particular, methylation of both the *TAp73* promoter (P1) and the *deltaNp73* promoter (P2) was higher in tumors compared to normal tissues, in ER-positive compared to ER-negative tumors (Figure 7A), and in luminal A and B compared to basal tumor subtype (Figure 7B). Thus, lower methylation levels were associated with less differentiated ER-negative and basal breast tumors. In addition, we analyzed formalin fixed paraffin embedded (FFPE) breast tumor specimens from *BRCA1/2* mutation carriers and found relatively high methylation levels (nearly 80%) at the *deltaNp73* promoter (P2) in all tumors, regardless of their subtype (Figure 7C). These findings suggest that the *deltaNp73* promoter is gradually hypermethylated according to breast cancer differentiation, from basal ER-negative tumors to luminal B and to the most differentiated ER+ luminal A tumors. 

## 3. Discussion

The potential of various tissues to develop cancer depends on a complex interplay between the unique genetic and epigenetic context of the tissue and environmental factors [25,26], as well as the number of cumulative stem-cell divisions through differentiation [27] and age-dependent aberrant DNA methylation [28]. Tissue-specific predisposition to cancer is particularly evident in hereditary cancer syndromes, where a germline mutation in master genes like Retinoblastoma (*Rb*), Mismatch Repair genes (MMR), or BRCA promotes cancer in certain tissues but not in others. However, the mechanisms underlying this phenomenon are still unclear [29]. In this work, we disclosed a breast-specific mode of deltaNp73 regulation, which might promote susceptibility to cancer specifically in *BRCA*-mutated breast epithelial cells (HMECs). Thus, we showed that the promoter of *deltaNp73* was unmethylated in normal HMECs but methylated in various other epithelial cell types. Accordingly, expression of deltaNp73 was markedly induced by DNA damage specifically in normal HMECs and protected them from ensuing cell death. A similar effect has been shown in primary mouse embryonic fibroblasts, where overexpression of exogenous deltaNp73 conferred resistance to spontaneous replicative senescence [19,30], which additionally caused cell immortalization. Increased genomic instability due to impaired response to replication stress was recently reported in BRCA1-haplo-insufficient HMECs [10,11] and some of these disorders were tissue or cell type-specific [12]. Here, we suggest that induction of deltaNp73 by replicative stress might rescue *BRCA1*-mut HMECs from senescence and contribute to genomic instability. More work should be done to reveal this point. Notably, the promoter of *deltaNp73* was highly methylated in normal ovarian and fallopian tube epithelium, therefore, deltaNp73 may not play a role in ovarian cancer susceptibility of *BRCA* mutation carriers.

Further, knockdown of BRCA1 in MCF10A cells led to augmentation of both deltaNp73 expression and its cell death-protective effect upon DNA damage. Moreover, overexpression of exogenous deltaNp73 increased cell viability of BRCA1-KD MCF10A but not of BRCA1-wt cells, following DNA damage. This suggests that deltaNp73 may play a unique role in BRCA1-deficient cells to inhibit DNA damage-induced apoptosis. A possible mediator between deltaNp73 and BRCA1 could be the DNA repair modulator 53BP1, which promotes premature senescence and apoptosis in BRCA1-deficient mice [31], interferes with homologous recombination while promoting non-homologous end-joining [32], and antagonizes BRCA1 during stalled forks restart [33]. Moreover, it was shown that 53BP1 interacted with deltaNp73, co-localized to the site of DNA damage, and inhibited the ATM-p53 pathway [22]. Of note, in our MCF10A model, Flag-deltaNp73 did not co-precipitate with 53BP1 or BRCA1 (Appendix A). However, these proteins may collaborate through indirect interactions, yet to be revealed. 

Unexpectedly, induction of deltaNp73 by DNA damage was associated with marked decrease in both BRCA1 and BRCA2 expression in HMECs and in MCF10A. This could be mediated by p53 transcriptional activation of deltaNp73 [34] and repression of BRCA1 [35,36]. Additionally, it was reported that induction of DNA damage by endogenous toxins caused depletion of BRCA2 and triggered spontaneous mutagenesis during DNA replication in immortalized breast epithelial cells with heterozygous mutation in *BRCA2* [37]. In line with these findings, we suggest that induction of deltaNp73 in *BRCA*-mutant HMECs accompanied by reduction in functional BRCA1/2 below a critical level could increase genomic instability and promote malignant transformation (graphic model, graphical abstract). 

Finally, unlike normal breast epithelium, the promoter of *deltaNp73* was hypermethylated in breast carcinomas, and the level of methylation varied across tumor subtypes. This is consistent with previous reports indicating that differentially methylated regions among various normal tissues are often also differentially methylated between normal tissues and their corresponding cancers [38]. DNA methylation is altered in cancer through multiple potential mechanisms including clonal selection [13]. This indicates that deltaNp73 may not play a major role beyond malignant transformation in breast tissues. However, additional work should clarify this point as variable expression of deltaNp73 protein was shown in breast tumors, and high expression levels correlated with worse prognosis [39].

In summary: DeltaNp73 may play a role in breast cancer susceptibility of *BRCA1* mutation carriers. In the future, it may evolve as a target for preventive therapy. 

## 4. Materials and Methods 

### 4.1. Primary Tissues, Primary Cells, and Cell Lines 

All tissue samples were collected and handled in compliance with the Institutional Review Board (IRB) constraints. Epithelial layers of human normal tissues were obtained from fresh surgical specimens. Healthy donors contributed white blood cells. Fresh normal breast tissues were isolated from prophylactic mastectomies in *BRCA1* mutation carriers and from reduction mammoplasties in non-carriers. Breast organoids (epithelial ducts and lobules) and mammary epithelial cells (HMECs) were enriched and cultured in specific growth medium according to [40]. Ovarian epithelial cells were purified and cultured according to [41]. Breast fibroblasts were isolated when occasionally exceeding epithelial cells in culture [40]. Human primary small airways epithelial cells were from PromoCell (GmbH, Heidelberg, Germany) and primary human prostate, colonic, and renal epithelial cells were from ScienceCell (Carlsbad, CA, USA). Cells of commercial origins were cultured with specific growth media according to the supplier’s instructions. MCF10A cell line was from ATCC (Manassas, VA, USA) and was authenticated by STR sequencing (BCF, Technion, Haifa, Israel). The cells were treated with Doxorubicin (Sigma, Merck, Darmstadt, Germany), Cisplatin (Teva, Nethanya, Israel), or were X-ray irradiated (Polaris sc-500 series II) at a dose rate of 100 cGy/min. All reagents for treatments were diluted in complete specific growth medium. Cell cultures were incubated in 5% CO_2_ and 70% humidity at 37 °C. 

### 4.2. DNA Methylation 

Genome wide DNA methylation analysis by Illumina 27K methylation array was previously described in [14]. DNA methylation analysis by quantitative methylation-specific PCR (q-MSP) including q-MSP primer sequences and Illumina probes sequences are provided in the Appendix A. 

### 4.3. Quantitative RT-PCR 

Total RNA was isolated from fresh tissues or cells using the SV Total RNA isolation kit (Promega, Madison, WI, USA). SuperScript III First-Strand cDNA Synthesis kit and Absolute Blue SYBR Green ROX mix (Thermo Fisher Scientific Inc., Waltham, MA USA) were used for RT-qPCR. PCR amplification (Rotor Gene 6000, Corbett) included 35 cycles of 5 s at 95 °C followed by 30 s at 60 °C. Primer sequences are provided in the Appendix A.

### 4.4. siRNA/shRNA Knockdown

Custom designed siRNAs specific for *deltaNp73* based on NCBI reference sequence NM_001126240.2 and non-target control siRNAs were from Invitrogen (Thermo Fisher Scientific Inc., Waltham, MA, USA). The design of siRNAs and the oligonucleotide sequences are provided in the Appendix A. siRNA was transfected using lipofectamine-RNAiMax (Invitrogen, Thermo Fisher Scientific Inc., Waltham, MA, USA) according to the supplier’s reverse transfection protocol. A transfection reaction in a 1 × 96-well included 0.2 µL siRNA (final concentration 20 nM) mixed with 0.2µL lipofectamine and 20 µL Opti-MEM medium (Gibco, Thermo Fisher Scientific Inc., Waltham, MA, USA). In total, 5000 cells in 0.1 mL growth medium were then added to the transfection mix and the cells were incubated for 24 h before treatment. ShRNA knockdown of BRCA1 used lentiviral vectors (pLV-mCherry-Neo-U), custom designed and cloned by VectorBuilder, Chicago, IL, USA. The shBRCA1 sequences were based on NCBI reference sequence NM_007294.3, provided in the Appendix A. Lentiviral particles were produced at the vector core facility, Tel-Aviv University, according to the protocol in [42]. For viral infection, 50,000 MCF10A cells in 1 mL complete growth medium were mixed with 150 μL of 10^6^ TU/mL viral particles and 8 μg/mL polybrene. Following 24 h of incubation, the viral particles were washed and the cells were incubated for 10 days in complete growth medium containing 100 μg/mL G418 (Sigma, Merck, Darmstadt, Germany) for positive selection.

### 4.5. DeltaNp73 Overexpression 

The deltaNp73 mRNA (NCBI Reference Sequence: NM_001126240.2) was cloned into lentiviral vector (pLV-Hygro-ΔNp73(ORF)-3xFLAG-T2A-EGFP) together with the co-vector (pLV-Bsd-EF1A-Tet3G), which allowed inducible expression by doxycycline. The vectors were custom designed and provided by VectorBuilder, Chicago, IL, USA. The vectors were packed in lentiviral particles and infected MCF10A cells as described above. The cells were selected for positivity by resistance to 50 μg/mL hygromycin B (Sigma-Merck, Darmstadt, Germany) and 10 μg/mL of blasticidine (BioVision Inc., Milpitas, CA, USA). DeltaNp73 induction used 0.01–1 μg/mL Doxycycline (Sigma-Merck, Darmstadt, Germany). 

### 4.6. Western Blot and Immunoprecipitation

Protein extracts from MCF10A cells or HMECs were prepared using 300 μL RIPA buffer supplemented with protease inhibitors cocktail (Roche diagnostics GmbH, Mannheim, Germany) per 1−3 × 10^6^ cells. Lysis was incubated on ice for 30 min and centrifuged for 10 min at 14,000× *g*, 4 °C. Bradford protein quantification used the Bio-Rad protein assay reagent (Bio-Rad, Hercules, CA, USA). Equal amounts of protein extracts (40−80 μg per lane) in Laemmli buffer were loaded on a gradient 4–12% polyacrylamide gel and transferred to nitrocellulose membrane. For immunoprecipitation, protein lysates (500 μg) were incubated with 3 μg of specific antibodies for 16 h at 4 °C, followed by 1 h incubation with Protein G magnetic beads (Dynabeads, Invitrogen, Thermo Fisher Scientific Inc., Waltham, MA, USA). The magnetic beads were washed 3× with TBS-T and eluted in Laemmli buffer. Primary antibodies for Western blots included anti-BRCA1(C−20) and anti-beta-Actin (clone C4, Santa Cruz, Dallas, TX, USA), anti-FLAG (Pierce, Thermo Fisher Scientific Inc., Waltham, MA, USA), and rabbit monoclonal anti-p73 recognizing the deltaNp73 isoform (clone D3G10, Cell Signaling Technologies, Danvers, MA, USA). Detection used ECL from Amersham (GE Helthcare, Merck, Darmstadt, Germany) and MicroChemi Imager (DNR Bio-Imaging Systems, Neve Yamin, Israel). Densitometry used the ImageJ software (ImageJ, U. S. National Institutes of Health, Bethesda, MD, USA, https://imagej.nih.gov/ij/). 

### 4.7. Cell Viability

MCF10A or HMECs were plated at 60–70% confluence in a 96-well culture dish (5 × 10^3^ cells in 0.1 mL complete growth medium per well), incubated over night for attachment, and treated with the indicated DNA damaging agents for 24−48 h. For cell viability assay, 0.05 mL of XTT reagent (Cell proliferation kit, Biological Industries, Beit Haemek, Israel) were added to each well. The colorimetric reaction was developed within 2–4 h incubation at 37 °C and absorbance was measured at a wavelength of 450–500 nm using a plate reader (Multiskan Ex, Thermo Fisher Scientific Inc., Waltham, MA, USA).

### 4.8. TCGA Data Analysis

TCGA methylation data were accessed through the Broad Firehose TCGA website (https://gdac.broadinstitute.org/). Data analysis was performed using the R Statistical Environment using Bioconductor packages and custom scripts. Annotations for Illumina Infinium 450K Human Methylation Microarray were accessed through IlluminaHumanMethylation450kanno.ilmn12.hg19 library. The PAM50 status was predicted using the package genefu and the pam50.robust model on RNA-seq data. Promoter analysis for *TAp73* (P1) and *deltaNp73* (P2) included Illumina probes as specified in the Appendix A. *p*-values for the differences in probe average beta-values between the indicated groups were calculated by Welch Two Sample t-test.

### 4.9. Statistical Analysis

Quantitative data are presented as mean ± standard error of the mean (S.E.M). *p*-values between groups were calculated by Student’s *t*-test (two-tailed), using Excel 2016. Unless noted otherwise, the results report the mean of three independent experiments. A linear mixed model, accounting for treatment and cisplatin as fixed factors, and sample ID as a random factor, analyzed significance of the effect of siRNA-deltaNp73 on cell viability in response to cisplatin, relative to siRNA-control. The mixed model analyses were done using R (v. 3.6.1) (R Foundation for Statistical Computing, Vienna, Austria. URL http://www.R-project.org/) and the packages ‘lme4’ and ‘lmerTest’.

### 4.10. Ethical Approval

The use of human tissue specimens in this study was approved by the Shamir (Assaf Harofeh) Medical Center Institutional Review Board, authorized by the Israeli Ministry of Health (Institutional Helsinki committee), reference No. 35/10*4 and 0072-17-ASF/20173814. All samples were obtained after explanation to the patient or healthy donors and after his/her written and signed informed consent.

## Figures and Tables

**Figure 1 cancers-12-02367-f001:**
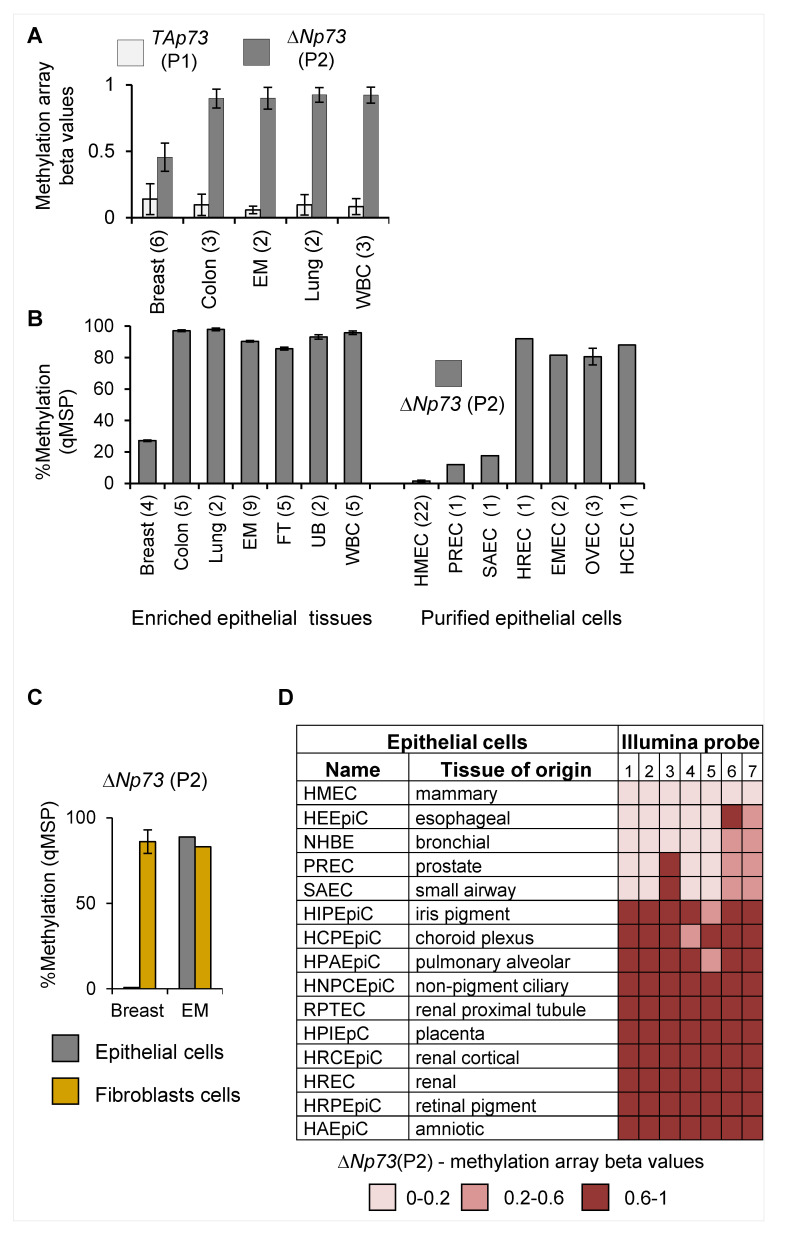
Tissue-specific DNA methylation of the *deltaNp73* promoter (P2) in normal human epithelial tissues and primary cells. (**A**) DNA methylation of the *TP73* promoters (P1 and P2) analyzed by Illumina methyl27K array. Data represent average beta values from 0 to 1 (non-methylated to fully methylated) ± S.E.M. (**B**) DNA methylation of the *deltaNp73* promoter (P2) measured by quantitative methylation-specific PCR (q-MSP) in various normal human epithelial tissues and purified primary epithelial cells. Data represent mean ± S.E.M. (**C**) Differential DNA methylation at the *deltaNp73* promoter (P2) in epithelial cells and in matched fibroblasts from breast (*n* = 3) or endometrial (EM, *n* = 1) tissues of the same individual. (**D**) DNA methylation of the *deltaNp73* (P2) promoter in various normal human epithelial cells. Data were recruited from the ENCODE/HAIB, UCSC Genome Browser on human Feb. 2009 (GRCh37/hg19) assembly and represent beta values of 7 of Illumina methyl450K array probes encompassing the promoter region. Tissue abbreviations: EM—endometrium; FT—fallopian tube; UB—urinary bladder; WBC—white blood cells. Epithelial cells: HMEC—mammary; PREC—prostate; SAEC—small airways; HREC—renal; EMEC—endometrial; OVEC—ovarian; HCEC—colon. (*n*) = number of tissue samples obtained from different individuals.

**Figure 2 cancers-12-02367-f002:**
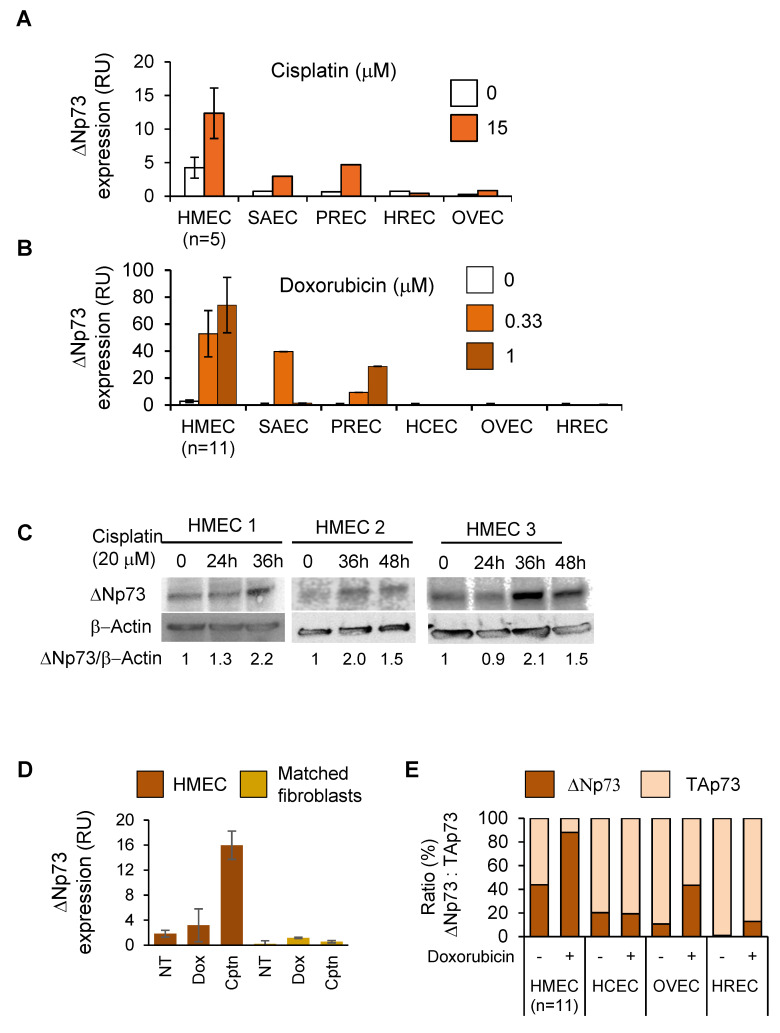
Tissue-specific induction of deltaNp73 by DNA damage in human epithelial cells. (**A**) Basal expression and induction of deltaNp73 (mRNA) following exposure to cisplatin or (**B**) doxorubicin for 24 h in HMECs as compared to other types of normal epithelial cells. (**C**) DeltaNp73 protein expression increased in HMECs from three individuals following exposure to cisplatin for the indicated time points. (**D**) DeltaNp73 expression (mRNA) in HMECs and in matched fibroblasts originated from the same breast tissue (*n*= two individuals) following treatment with 0.33 μM doxorubicin (doxo) or 20 μM cisplatin (cptn) for 24 h. (**E**) The ratio of deltaNp73:TAp73 expression (mRNA) was greater in HMECs (*n* = 11 individuals) as compared to other epithelial cells (*n* = 1 individual) following induction with 1 μM doxorubicin for 24 h. Epithelial cells: HMEC—mammary; SAEC—small airways; HREC—renal; PREC—prostate; OVEC—ovary; HCEC—colon. Gene expression (mRNA) was measured by RT-qPCR and normalized to mRNA expression of the housekeeping gene, GAPDH (RU = 2^ΔCT^ × 10^4^). Error bars represent S.E.M for *n* > 1.

**Figure 3 cancers-12-02367-f003:**
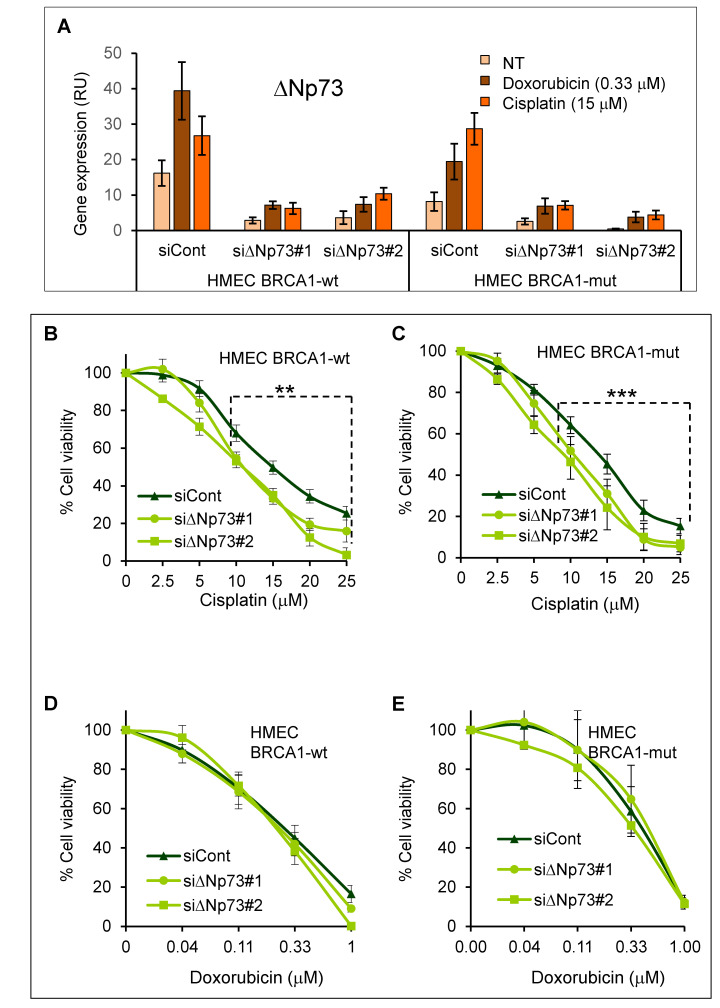
DeltaNp73 protected normal mammary epithelial cells (HMECs) from cisplatin-induced cell death. HMECs from healthy women with wild-type *BRCA1* (*BRCA1*-wt, *n* = 7 individuals) and from *BRCA1* mutation carriers (*BRCA1*-mut, *n* = 5 individuals) were transfected with non-target siRNA (siControl) and with two different siRNAs targeting deltaNp73 (siΔNp73#1, *n*-wt = 7; *n*-mut = 4 and siΔNp73#2, *n*-wt = 5; *n*-mut = 4 individuals). Twenty-four hours after transfection the cells were exposed to increasing doses of cisplatin or doxorubicin for 48 h. (**A**) Expression of deltaNp73 (mRNA) following induction with DNA damage and inhibition by siRNA. Data of RT-qPCR were normalized to GAPDH (RU = 2^ΔCT^ × 10^4^). (**B**–**E**) Cell viability was measured by tetrazolium salt (XTT)-based colorimetric assay and normalized to the measurement of the untreated cells as 100%. Each data point on graphs represent the average of all individuals as indicated above (The transfection experiment for each individual was repeated twice and XTT was done in triplicate) ± S.E.M. The differences between siControl and siΔNp73#1&2 at the range of 10−25 μM cisplatin (b, c) were calculated by linear mixed model analysis as described in Appendix A. Accordingly, *p*-values were ** *p* = 0.0085 for *BRCA1*-wt and *** *p* = 0.0004 for *BRCA1*-mut HMECs.

**Figure 4 cancers-12-02367-f004:**
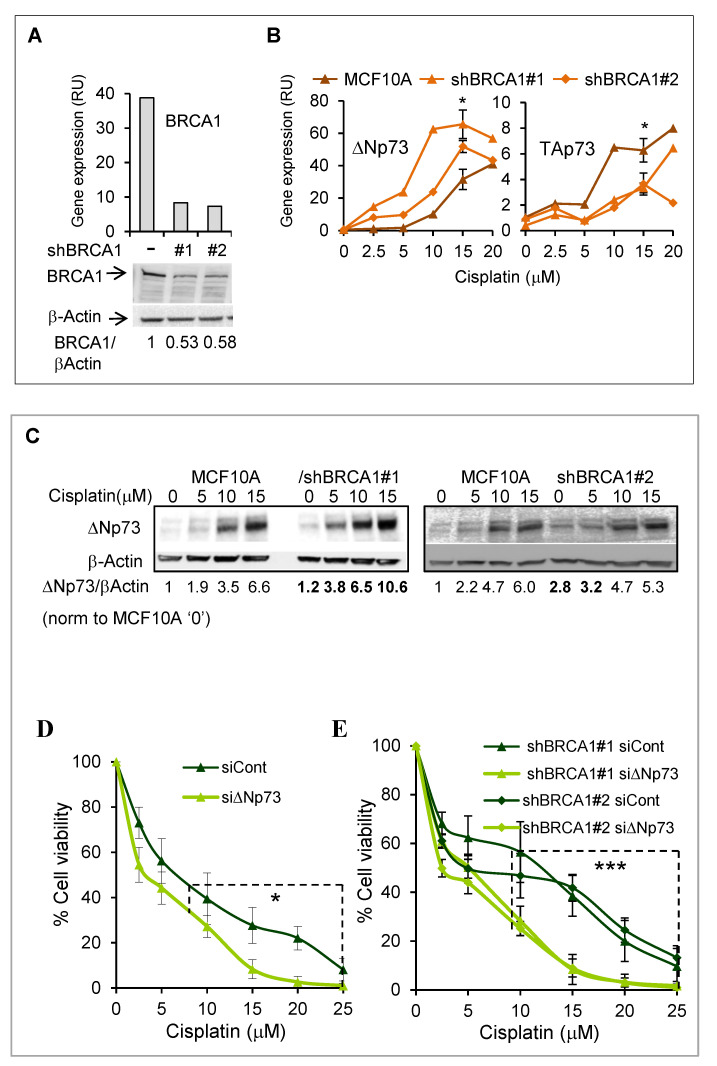
Induction of deltaNp73 and its protective effect on cell viability were augmented in BRCA1 knocked-down MCF10A following treatment with cisplatin. Stable knockdown of BRCA1 (BRCA1-KD) in MCF10A was performed by two different shRNAs (shBRCA1#1 & #2). (**A**) BRCA1 mRNA (upper panel) and protein (lower panel) are shown for BRCA1-wt cells and the BRCA-KD clones. (**B**) Induction of deltaNp73 mRNA by cisplatin was greater in the BRCA1-KD than in the parental MCF10A cells (left), while the induction of TAp73 was reciprocally suppressed in the BRCA1-KD cells (right). The data point of 15 μM cisplatin represents average of three independent experiments with error bars of S.E.M. The differences in gene expression between the parental MCF10A cells and the average of the two BRCA1-KD clones were calculated by paired, 2-tail Student’s t-test (* *p* < 0.05). (**C**) Induction of the deltaNp73 protein by cisplatin was augmented in BRCA1-KD (#1) compared to the parental MCF10A. Proteins were analyzed by Western blot and signal intensity was quantified by ImageJ software. DeltaNp73 quantification was normalized to beta-Actin in each lane and shown as folds of the value measured for the untreated parental MCF10A. (**D**,**E**) MCF10A and MCF10A/BRCA1-KD cells were transfected with non-target siRNA (siControl) or with siRNA targeting deltaNp73 (siΔNp73) and exposed to cisplatin for 48 h. Appendix A shows the assessment of deltaNp73 expression in these cells. Cell viability was measured by tetrazolium salt (XTT)-based colorimetric assay and normalized to the measurement of the untreated cells as 100%. Linear mixed model statistical analysis compared siControl and siΔNp73 datasets at the range of 10−25 μM cisplatin (Appendix A) and showed that this effect was more significant in BRCA1-KD (*** *p* = 0.000078 for shBRCA1#1 and *** *p* = 0.00014 for shBRCA1#2) than in the BRCA1-wt (* *p* = 0.0379) MCF10A. Error bars represent the S.E.M for three independent experiments.

**Figure 5 cancers-12-02367-f005:**
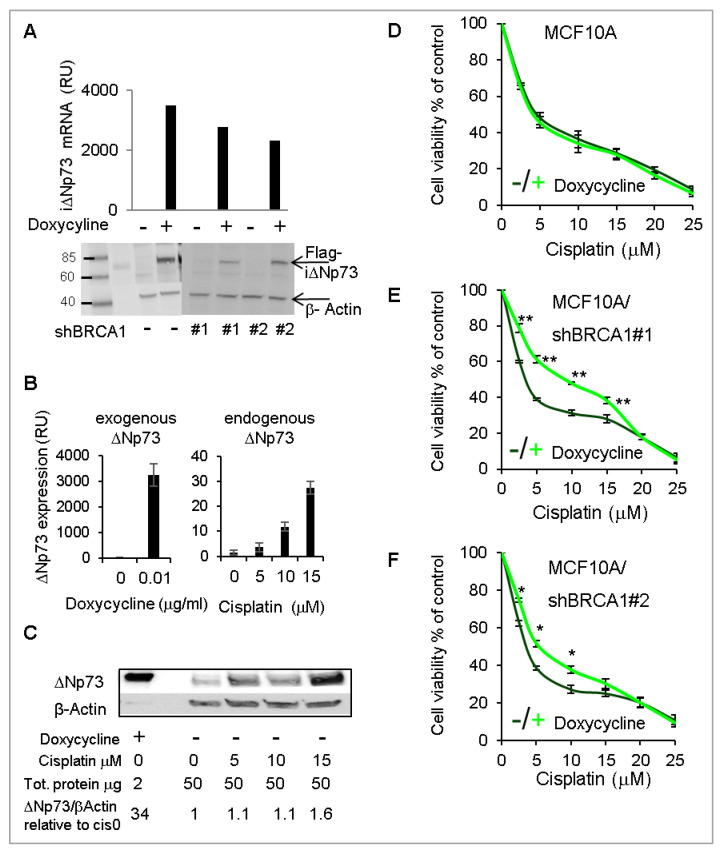
Over expression of deltaNp73 protected BRCA1-KD MCF10A from cisplatin-induced cell death, while the BRCA1-wt MCF10A were unaffected. Exogenous Flag-deltaNp73 was stably expressed in MCF10A and in the BRCA1-KD clones under the control of a doxycycline-inducible promoter. (**A**) Induction of flag-deltaNp73 (iΔNp73) by 0.01 μg/mL doxycycline for 24 h is shown by mRNA (RT-qPCR) and protein (Western blot using Flag antibodies). Corresponding expression of BRCA1 in these clones is shown in Appendix A. (**B**,**C**) Comparison of iΔNp73 mRNA and protein induced in MCF10A by 0.01 μg/mL doxycycline for 24 h to the endogenous deltaNp73 induced by cisplatin in the same cells. The data were quantified by ImageJ software and deltaNp73 values were normalized beta-Actin and calculated as relative folds of deltaNp73 in the untreated MCF10A (no doxycycline, no cisplatin). (**D**–**F**) MCF10A shown in (**A**) were exposed to 10 ng/mL doxycycline (or not), and simultaneously treated with increasing doses of cisplatin for 48 h. Cell viability was measured by XTT. Data represent the mean ± S.E.M of three independent experiments. Each experiment was done in triplicate. *p*-values were calculated by paired, 2-tail Student’s t-test, for each concentration of cisplatin separately. Stars are indicative of *p*-values: * *p* < 0.05, ** *p* < 0.01.

**Figure 6 cancers-12-02367-f006:**
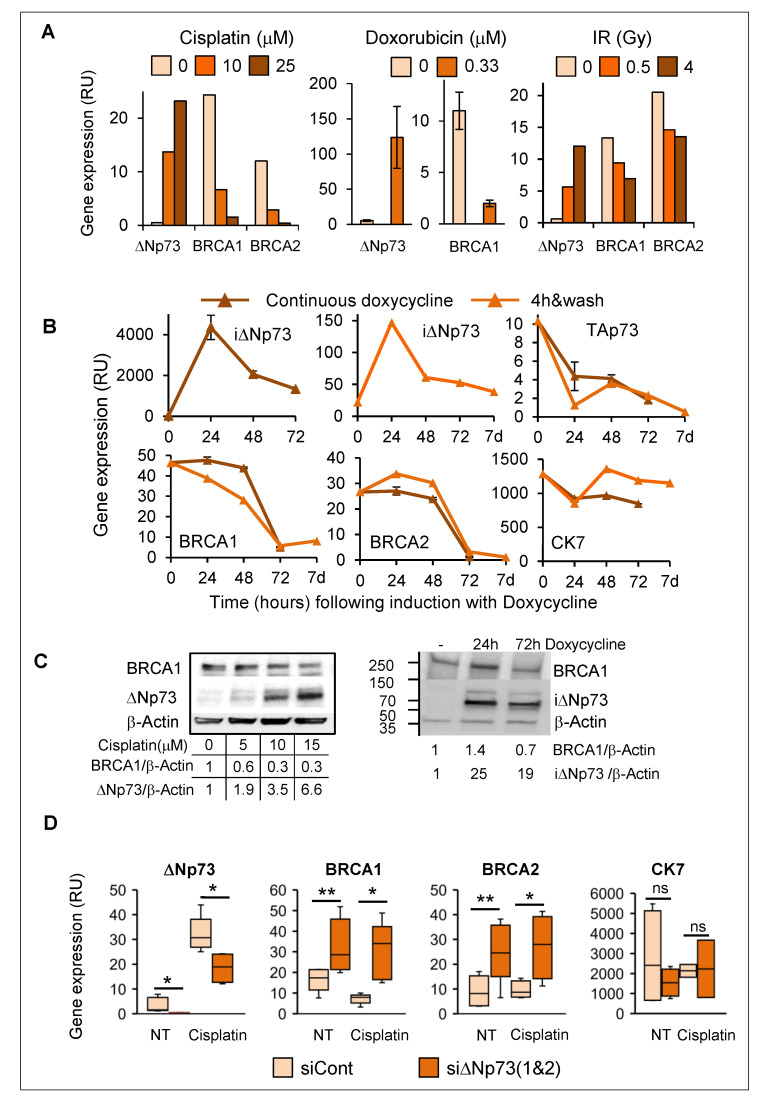
Induction of deltaNp73 corresponded with a decrease in BRCA1 and BRCA2 gene expression. (**A**) DeltaNp73 expression (mRNA) was induced, while BRCA1/2 expression was reduced in MCF10A following exposure to various DNA damaging agents for 24 h. Error bars represent the mean ± S.E.M when the experiment was repeated three times. (**B**) Overexpression of exogenous deltaNp73 in MCF10A was accompanied by reduction in BRCA1 and BRCA2 expression 72 h post exposure to 0.01 μg/mL doxycycline. TAp73 was reciprocally reduced while CK7 remained at high levels. (**C**) BRCA1 protein was reduced as deltaNp73 increased in cisplatin treated MCF10A (left) and 72 h post induction of MCF10A with doxycycline. Western blot data were quantified by ImageJ software and deltaNp73 values were normalized beta-Actin and calculated as relative folds of deltaNp73 in the untreated MCF10A (no cisplatin, no doxycycline). (**D**) Inhibition of deltaNp73 in MCF10A by siRNA was associated with increased BRCA1 and BRCA2 gene expression (mRNA). MCF10A were transfected with siControl or siΔNp73, incubated for 24 h, and then, treated with 15 μM cisplatin for additional 24 h before RNA extraction. Gene expression was measured by RT-qPCR and normalized to GAPDH (RU = 2^ΔCT^ × 10^4^). Box plot lower and upper limits represent the Q1 and Q3 values and the lines inside the boxes represent the mean. Error bars represent S.E.M (*n* = 5 independent experiments). Statistical significance of the difference between siControl to sidNp73 was analyzed by non-paired, 2-tail Student’s *t*-test (* *p* < 0.05, ** *p* < 0.01).

**Figure 7 cancers-12-02367-f007:**
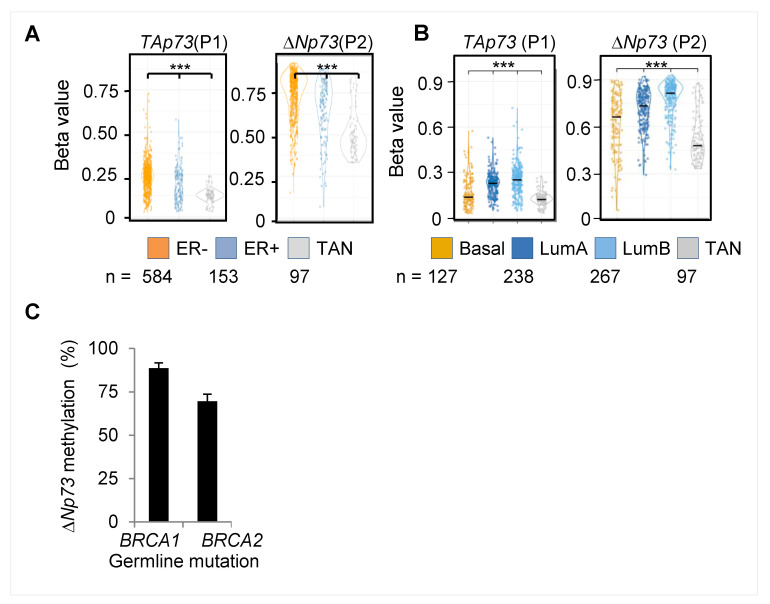
DNA methylation at the *TP73* gene in breast cancers. (**A**,**B**) TCGA analysis shows distribution of average methylation groups for all Illumina probes located at the P1 or the P2 promoter regions among (**A**) ER+ and ER- breast tumors or (**B**) PAM50 breast tumor subtypes. *** represent *p*-values < 0.001 by Welch two-sample t-test. (**C**) q-MSP analysis of the *deltaNp73* promoter (P2) in breast tumors of *BRCA1* (*n* = 16) and *BRCA2* (*n* = 6) germline mutation carriers. Columns represent the mean ± S.E.M. **ER**- estrogen receptor, Lum–luminal, TAN–tumor adjacent normal tissue

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
