# Peer review of "Breast-Specific Epigenetic Regulation of DeltaNp73 and Its Role in DNA-Damage-Response of BRCA1-Mutated Human Mammary Epithelial Cells"

_cancers, 2020, doi:10.3390/cancers12092367_

Round 1

Reviewer 1 Report

This is an interesting original research work evaluating the role of breast-specific epigenetic regulation of deltaNp73 may facilitate in BRCA1-mutated human mammary epithelial cells cancer susceptibility. However, the following issues need to be discussed.

Majors:

  • Please further study deltaNp73 promoter methylation in peripheral blood cells of breast cancer patients to consider germline BRCA mutation and give more scientific strength to your findings.

Minors:

  • Consider to change title since it should not directly point out the final results of your research and uncommon acronyms should be spelled out (e.g: the role of breast-specific epigenetic regulation of deltaNp73 in BRCA1-mutated human mammary epithelial cells [HMECs])
  • Genes (but not proteins) must be typed in italics
  • Do consider to use the past tense when presenting your final results

Reviewer 2 Report

The authors show that the promoter of deltaNp73 is unmethylated in normal human breast epithelium and methylated in various other normal epithelial tissues and cell types. deltaNp73 was found to be markedly induced by DNA damage in human mammary epithelial cells (HMECs) and the induction of deltaNp73 protected HMECs from DNA damage-induced cell death,  and this effect was more substantial in HMECs from BRCA1 mutation carriers. Moreover, BRCA1 was knocked-down in MCF10A non-malignant breast epithelial cell line, both deltaNp73induction and its protective effect from cell death were augmented upon DNA damage. Interestingly, deltaNp73 induction also resulted in inhibition of BRCA1 and BRCA2 expression following DNA damage. Therefore, they propose a new mechanism that deltaNp73 acts as oncogenic gene in the background of BRCA1 mutation carrying breast cancer patients. This paper is well written and worth publishing in Cancers after minor English/formal verification and formatting it. It is also highly recommended to have a summary of this study as a new figure at the end to appeal/facilitate understanding of what they found in this paper.

-L172 Off note

-L401-407 Formatting error (Font and size)

-Did this study obtain local ethical committee approval for the use of BRCA1 mutant cells? The information should be provided.

Reviewer 3 Report

The paper is very good and well explained. I have no concern with the quality of the data presented. The paper makes a nice concise story and the data is supportive of the hypothesis and includes sufficient controls and statistical analysis. I have included a couple of editorial changes and a couple of questions for your consideration

What is happening with the other members of this family (p53, P63) under the conditions of this experiment? Do they follow a similar or different pattern of expression?

If there were 110 differently methylated regions in breast tissue, how was TP73/NP73 selected for further study?

Line 172 “Off note” I think you meant “Of note”

Line 174 – Include space after period, before “Furthermore”

Lines 410-407 – Some of the text is a different font size.

One thing I did not follow. If delta NP73 is hypomethylated in breast tissue and this permits its expression (lines 85-86) protecting the cells from apoptosis and permitting additional genomic instability and contributing to promoting cancer, why is it that it is more methylated in breast cancer tissue samples (section 2.7)?

Round 2

Reviewer 1 Report

Thank you for for implementing the flaws point by point.